# Robotics in Total Hip Arthroplasty: Current Concepts

**DOI:** 10.3390/jcm11226674

**Published:** 2022-11-10

**Authors:** Emily K. C. Bullock, Michael J. Brown, Gavin Clark, James G. A. Plant, William G. Blakeney

**Affiliations:** 1Department of Orthopaedic Surgery, Royal Perth Hospital, Wellington St., Perth, WA 6000, Australia; 2Department of Orthopaedic Surgery, St John of God Subiaco Hospital, 12 Salvado Road, Subiaco, WA 6008, Australia

**Keywords:** total hip arthroplasty, robotics, functional outcomes, radiological outcomes, surgical technique

## Abstract

This current concepts article reviews the literature pertaining to the use of robot-assisted systems in total hip arthroplasty (THA). The bulk of the literature is regarding the MAKO (currently the most used system worldwide) and the historic ROBODOC robotic systems. There is a paucity of literature available on other systems, with several still in pilot-phase development. Whilst the evidence shows improved radiological outcomes with robotic THA, functional outcomes are equivocal between conventional and robotic techniques. Acceptance of robotic THA worldwide is limited by its accessibility including cost, and by already exceptional results with the conventional technique. It is, however, a rapidly developing area of orthopaedic surgery. This article discusses the history of robotics in THA, current surgical techniques, functional and radiological outcomes, and ongoing avenues for development.

## 1. Introduction

The first successful total hip arthroplasty was performed in 1951 by McKee and Farrar [1]. Initial attempts at THA were troubled by inadequate fixation methods in the femur and pelvis, in addition to significant soft tissue reactions secondary to poorly performing bearing materials, with resultant poor survivorship. THA was revolutionised in 1971 with the development of the low friction arthroplasty by Sir John Charnley [2]. The results of this implant demonstrated survivorship in the region of 80% at 25 years and led to some authors heralding THA “The operation of the century” [3]. Since Charnley’s early success, the outcomes for THA have continued to improve. Currently, the benchmark for THA revision in the Australian Orthopaedic Association’s Joint Replacement Registry (AOANJRR) is 4.4% at 10 years and 6.5% at 15 years [4]. Improvements have undoubtedly resulted from reduced wear rates associated with improved bearing materials, increasingly robust fixation methods and modularity within implants allowing the surgeon to recreate patient anatomy with greater accuracy. More recently, increasing focus has turned to the use of robotics in THA in the hope that it may provide a gateway to further progress. Proponents of robotics hope that it can lead to more individualised implant positioning, better precision, and fewer surgical errors, which in turn will result in improved patient outcomes with lower revision rates. Whilst the use of robotics within other surgical specialties quickly became commonplace, the uptake has been slower within orthopaedics. The reasons for this are likely multifactorial, but may be related to the significant costs, the learning curves associated with the technique and the already impressive results associated with conventional techniques. This article discusses the history of robotics in THA, its current role and use, outcomes, and ongoing avenues for development.

## 2. History

It is widely accepted that component positioning in THA is crucial to patient function and implant survival. Component malpositioning can increase wear rates in THA, shortening the longevity of the implant [5]. Inaccurate positioning can also lead to dislocation, impingement or periprosthetic fracture [6]. The use of robotics in THA has evolved over the past 30 years, attempting to reduce human error in component positioning and thus to improve outcomes in this widely adopted procedure.

The concept of automated arthroplasty was first raised in 1986 by Drs Howard Paul and William Barger. Their system, known as ROBODOC Surgical Assistant Systems (Sacramento, CA, USA), was an autonomous system that required the operator to input preoperative planning information, and would then perform the procedure without direct surgeon guidance. After placing navigational pins on the femur, a computed tomography (CT) scan was taken and loaded onto a computer workstation called ORTHODOC. This workstation then produced a three-dimensional model of the bony anatomy and was used to select appropriately sized components for accurate insertion in the respective patient [7]. With the patient rigidly fixed with bone pins to maintain a stable working field, ROBODOC then automatically carries out preparation of the bony surfaces to match the preoperative surgical plan [8].

Following a canine trial and a feasibility study, a multicentre clinical trial was undertaken in 1993 involving 300 patients, of which 150 received a robotic, and 150 received a conventional THA [9]. Although there was no significant difference in function or hospital length of stay between treatment arms, the ROBODOC group had significantly improved femoral component positioning [10]. By the late 1990s, over 800 patients worldwide had received a ROBODOC-assisted robotic THA and by 2017, over 17,000 [7,11]. The ORTHODOC system remains in use today, although at various stages since its development, concerns have been raised about the rates of soft tissue injury, periprosthetic fracture and a relatively high conversion rate to conventional THA intraoperatively [12].

In the intervening years, several other robotic-assisted modalities have been developed and introduced into clinical practice. The CASPAR (Rastatt, Germany) robotic-assisted system was introduced in Germany in the 1990s. Like the ROBODOC system, it was also an autonomous/active robotic-assisted system; however, it is not in current use today after early studies showed unacceptable intra-operative complication rates and poor post-operative functional outcomes when compared to conventional THA [13]. 

Problems associated with active robot systems, given their lack of surgeon input, led to the development of semi-active systems. The ACROBOT (The Acrobot Co. Ltd., London, UK) was initially developed for use in hip resurfacing and the system involved the surgeon moving a robotic arm which constrained within a pre-planned field to prevent diversion from the preoperative plan. This technology was acquired by Stryker Ltd. (Kalamazoo, MI, USA) after the development of the MAKO robotic-arm-assisted system, to settle a patent infringement in 2013 [14]. The MAKO system functions in a similar fashion as a haptic/semi-active system where the surgeon can operate the robotic arm manually within predetermined system constraints for placement of the acetabular component of the THA. It provides visual, tactile (haptic), and auditory prompts, minimising deviation from the pre-operative plan [15] (Figure 1). MAKO Surgical Corp. was founded in 2004, with the initial area of advancement being in partial knee arthroplasty. The first MAKO robotic-arm-assisted THA was undertaken in October 2010 with commercial launch in 2011. MAKO Surgical Corp. was acquired by Stryker in 2013. In 2015, they released their third-generation THA, which enabled surgeons to use Stryker implants on the MAKO platform, and this model received FDA-approval in the same year. This system uses a range of implants and is also based on a pre-operative CT that creates three-dimensional images of patient anatomy and can be applied to all approaches to THA [16]. The MAKO robotic-arm-assisted system is the most widely adopted robotic arthroplasty system worldwide and currently dominates the market, but other companies have developed semi-active systems of their own, including the ROSA system (Zimmer-Biomet, Warsaw, Indiana), VELYS (DePuy Synthes, Rayham, MA, USA), and CORI (Smith & Nephew, Watford, UK).

Where the ROSA system differs is that it is explicitly used for the direct anterior surgical approach to the hip. The ROSA does not use pre-operative CT scanning or bone tracker pins for intra-operative navigation, and instead uses intra-operative fluoroscopic guidance. Their surgical planning tool uses preoperative X-rays as a guide [17] (Figure 2).

Whilst many other commercially available orthopaedic robotic systems are not currently utilised for hip replacements, the CORI system has been used for THA in pilot cases with further developments ongoing.

## 3. Current Role

The implementation of robotic THA systems worldwide has increased exponentially in recent years. A multidisciplinary approach to research and system implementation has led to this increase, most notably since 2014. The predominant body of evidence is centred on the three robotic hip systems, MAKO, ROBODOC, and CASPAR, which are manufactured in the United States and Germany [18]. Whilst this paper will primarily review the MAKO robotic-arm-assisted system given its current dominance in robotic hip arthroplasty, it will make brief mention of other systems where their processes markedly differ.

### Stages of Robotic Total Hip Arthroplasty


*(1) Preoperative planning*


In THA, planning involves adequate imaging for understanding the relevant patient anatomy, and to ensure the most appropriate implant and its relative position is chosen for the patient. This phase is a prerequisite for robotic THA; however, it has been introduced for some manual systems as well. A CT scan of the pelvis and full-length femur is taken to provide cross-sectional information in axial, sagittal and coronal planes [19]. This is used to computer-generate a patient-specific 3D model of their native pelvic and femoral anatomy, which is then used to create a preoperative plan for implant size, type, and positioning. Bony landmarks are the basis upon which accurate implant positioning is determined, and are used to calculate cup position, inclination, version, and centre of rotation (COR), femoral stem size and head length, combined offset, and correction of any pre-existing leg length discrepancy (Figure 3 and Figure 4).

In contrast to MAKO, the ROBODOC system requires patients to undergo a separate procedure under local anaesthesia in an outpatient surgical setting within 24 h of the surgery to place locator pins in the femur (two in the epicondyles above the knee and one in the greater trochanter). The patient then has a CT scan which is then transferred to the ORTHODOC planning station. Implant position is determined by the surgeon on the scan and then this information is transferred to ROBODOC [8].

The ROSA system does not use a CT scan preoperatively but relies on plain radiographs, with antero-posterior and lateral standing and sitting views [20].

The importance of spinopelvic mobility to the success of THA surgery has been recognised and incorporated into contemporary preoperative planning systems. Spinopelvic mobility incorporates the complex relationship between the spine, pelvis, and hip. Patients with advanced arthritis may have abnormal spinopelvic alignment or sagittal imbalances through the flexion arc between sitting and standing [19]. Consequently, this cohort has higher rates of dislocation, especially in those with biological or surgical spinal fusions, due to malpositioning of the acetabular component [21,22,23]. In a conventional THA, it is difficult to conceptualise the spinopelvic relationship. Integration of spinopelvic parameters into robotic technology, however, can guide restoration of native pelvic kinematics [24,25].


*(2) Surgical preparation and approach*


The operative plan is displayed on a computer and positioned in front of the surgeon. The choice of surgical approach is surgeon-dependent and must be applicable to the robotic system in use. For example, with the MAKO system and posterior approach, the patient is in the lateral decubitus position, and as guided by the computer software prior to sterile preparation, the surgeon inserts navigation pins into the iliac crest of the operative hip with an aseptic technique (Figure 5).

The standard direct anterior approach is slightly modified for a robotic procedure. The patient is still supine; however, the incision is more oblique distally towards the thigh, to allow reduced muscle tension and exposure of the anterior trochanteric region [26]. The pins are placed into the thickest part of the contralateral iliac crest, and the pelvic attachment device is inserted on to the pins to connect the surgeon and software via the infrared camera as the acetabulum is registered [24,25]. Once this is complete, the joint is exposed without input from the robot.


*(3) Surgical technique*


(i) Registration process

Each bone that is to be prepared by the MAKO robot requires registration, and is fixed to either the robot itself, or a tracker. Once the femoral head has been dislocated, two further screws can be inserted into the proximal femur: one to hold the array for the infrared camera and one as a “checkpoint” for accuracy of bone registration [23] (Figure 6). The specific positions of the pins are determined by the surgical approach used [25]. Using the previously inserted navigation pins as a guide, 32 specific bony points are registered with a probe to allow the robot to match the patient’s anatomy with the preoperative CT scan. The same process is repeated for the acetabulum, with a pelvic check point screw inserted in the posterosuperior aspect of the acetabular rim, and 32 points are again registered [24] (Figure 7). If these screws are to come loose, values become inaccurate, and the registration process must be repeated [26].

For the ROSA system, once ensuring the C-Arm is accurately positioned over the pelvis, AP images of the hip and pelvis are taken. Landmarks for reference are then positioned at the teardrop, anterior pelvic brim, obturator foramina minor and major, lesser trochanter, medial and lateral aspects of proximal femoral shafts, and the centre of the femoral head [20].

For ROBODOC, after diagnostic checks, the surgeon orientates the robot to the femoral pins and directs the robotic arm to the femoral cavity [7].

(ii) Femoral preparation

Whilst the femur can use navigation assistance from the robot, the execution is manual, hence the majority of MAKO THA’s are performed with pelvic pins only, and femoral version is determined visually and confirmed robotically. Once registration is complete, the level of neck osteotomy is determined by again touching the probe to the bone (Figure 8). This line is marked with electrocautery or surgical marker on the proximal femur, the cut is made, and the femoral head resected [27]. Sequential broaches are used to prepare the femoral canal in line with the surgical technique for the chosen implant, with the final broach measuring the anteversion. The femoral broach array can be attached to the corresponding neck taper, and position assessed. Combined anteversion values can be altered once this final broach version is confirmed by changing the planned acetabular orientation. The femoral stem is inserted manually and seated with appropriate instrumentation [14,24,27].

The ROSA system requires the femoral canal to be prepared manually with conventional instrumentation. Once this is done, ROSA will progress to the trial step to measure leg length and offset discrepancies [20].

(iii) Acetabular preparation and placement

Once the surgeon has decided upon the final desired cup position, the MAKO system takes the planned position of the acetabular component and determines a haptic zone within which the reamer can move. This provides auditory, tactile, and on-screen feedback to ensure accuracy with the preoperative plan. A single reamer is utilised, and the robotic arm is manoeuvred within the zone to ream the predetermined area within the pelvis to allow accurate implantation of the acetabular component [14]. The component is maintained within 3 degrees of planned inclination and version by the robotic arm, whilst the surgeon impacts the shell, monitoring depth with optic tracking during the process [24,27] (Figure 9, Figure 10, Figure 11 and Figure 12). Final cup position is then assessed and has been shown to be reproducible and accurate.

In contrast, the ROSA system requires the acetabulum to be prepared and completed before the femur. The femoral head is resected, and the acetabulum is reamed manually according to the surgeon’s plan. Once the implant is positioned relative to the acetabulum, the robotic arm is connected, and the image is recalibrated with the same landmarks referenced again. The preoperative X-ray can be superimposed on the new images for comparison. The cup is moved into a desired orientation under fluoroscopic guidance, aiming for inclination and version within 2 degrees of the target values. Once the position is confirmed, the cup is impacted [20].

(iv) Intra-operative assessment of stability and position

With the MAKO system, once the femoral taper and head are positioned, the hip is reduced, and the stability of the prosthesis is assessed. The femoral array is placed on to the femoral screw, and the computer assesses leg length and offset values through motion, displaying the final values as compared to the preoperative plan [26] (Figure 13). The hip is moved through range to assess stability, and any adjustments to a shorter or longer head length can be made. The software must be updated if there is any deviation for the preoperative plan [27].

ROSA uses the same landmarks throughout to assess leg length and offset and calculates any discrepancy. Every fluoroscopic image must have the teardrop, proximal third of the femoral shaft, and lesser trochanter visible, and the acetabular component centred in the image to maintain accuracy. The reference landmarks and completion images are overlaid to make final comparisons [20].

## 4. Outcomes and Current Controversies

### 4.1. Radiological Outcomes

#### 4.1.1. Accuracy of Implant Placement

Accuracy of implant placement, as assessed radiographically, is a key point in the analysis of robotic THA outcomes. More accurate acetabular component positioning reduces the risk of dislocation and ultimately revision. To achieve this, surgeons use pre-operatively determined “safe zones”, as defined by either Lewinnek et al. (inclination 10–30 degrees; anteversion 5–25 degrees) or Callanan et al. (inclination 30–45 degrees; anteversion 5–25 degrees) [28,29,30].

Emara et al. described in their systematic review that robotic THA had superior acetabular cup positioning within both Lewinnek’s and Callanan’s safe zones in the 10 studies they reviewed [31]. Additionally, Chen et al., in a meta-analysis primarily assessing complications post-robotic THA, found more accurate acetabular cup placement in the robotic cohort, which they perceived to be advantageous towards less experienced surgeons, as the robotic systems allows the surgeon to assess cup placement intra-operatively, as well as improve accuracy in patients with an increased body mass index (BMI) [32].

#### 4.1.2. Heterotopic Ossification (HO)

HO is a post-THA finding that describes abnormal bone growth around soft tissues, conferring increased joint stiffness and reduced movement. Chen et al.’s meta-analysis found higher HO rates post-robotic THA; however, Han et al.’s meta-analysis (with the difference between the two being the inclusion of Honl et al.’s prospective study that demonstrated equivocal HO rates) found no significant difference between the robotic and conventional cohorts. It is important to note that all observed studies were based on the ROBODOC system [15,33,34]. Given HO is attributed to muscle trauma, it was initially expected that this should be lower in a robotic THA cohort. However, with a more accurate robotic-guided resection, the ROBODOC system requires greater soft tissue exposure for pin placement which may contribute.

#### 4.1.3. Leg Length Discrepancy (LLD)

LLD of varying degrees is relatively common post-THA and is one the leading causes of legal action against orthopaedic surgeons [35]. It is generally accepted that the patient will be cognisant of the discrepancy if shortening is >10 mm or lengthening is >6 mm [36]. Several studies have reported on the resulting LLD between conventional and robotic THA. Clement et al. showed significance in restoration of leg length in a robotic treatment arm [37]. In nine studies reviewed in their meta-analysis, Kumar et al. showed a statistically significant reduction in LLD in the robotic THA cohort [38]. Conversely, Domb et al., in a comparative analysis of 1980 hips managed with one of six surgical techniques, including robotic-guided anterior and posterior THA, conventional and navigation and fluoroscopic-guided THA, demonstrated rates of LLD to be comparable across all treatment arms, and within an acceptable range [39]. Emara et al. found in their meta-analysis robotic THA to have a significantly lower LLD across nine studies [31].

### 4.2. Functional Outcomes

Many studies have reported functional outcomes (including patient reported outcome measures [PROMs]) between robotic and conventional THA. Clement et al. assessed 120 patients (40 robotic and 80 conventional) with a mean follow-up of 10 months [37]. Their robotic cohort had significantly greater postoperative Oxford Hip Scores (OHS) and Forgotten Joint Scores (FJS), and the smaller standard deviation for these groups relative to the conventional group suggested a more reliable distribution of outcomes. No patients were “dissatisfied” by their robotic hip, whilst six patients from the conventional group were, although this was statistically insignificant [37]. Similarly, in a propensity score-matched study of 66 robotic and 66 conventional THAs followed-up over a minimum 5-year follow-up period, Domb et al. showed significantly higher Harris Hip scores (HHS), FJS and Veterans RAND 12 Physical (VR-12 Physical) for their robotic cohort [40].

Nishihara et al. assessed functional outcomes relative to stem implantation in 156 primary hips (78 robotically milled, 78 hand-rasped). Although there was no difference between the two cohorts in time to walking 500 metres with no walking aid, the robotic THA cohort did have a greater number of patients walking six blocks within 13 days of the operation, and this was statistically significant [41]. Additionally, whilst there was no difference in the Merle D’Aubigne hip score pre-operatively or at 3 months post-operatively, at 2 years the robotic cohort had a significant improvement. This suggests there may be some benefit in both the early post-operative period for rehabilitation and long term [41].

In contrast, Samuel et al.’s systematic review appraised 18 studies with a total of 2811 patients and overall found no significant difference in functional outcomes across a range of PROMs between the robotic and conventional THAs. Interestingly, when evaluating the MAKO and ROBODOC systems independently across PROMs, a majority of pooled analyses also found no significant difference [42]. Han et al. included 14 trials in their meta-analysis, of which 6 studies assessed functional outcomes across the HHS, Western Ontario and McMaster Universities (WOMAC) osteoarthritis index or the Merle d’Aubigne hip score. No study demonstrated significant difference in post-operative clinical outcomes between the robotic and conventional THA [33].

Overall, Chen et al. have shown that whilst there may be some medium-term studies that show the robotic THA cohort have improved functional scores at the 2–3 years post-operative mark, they are equivocal at 5 years [32].

### 4.3. Complications

#### 4.3.1. Infection

Prosthetic joint infection can be catastrophic, and minimisation of risk and management of infection is important when assessing outcomes pertaining to hip arthroplasty. Illgen et al. found no difference in infection rates between matched robotic and conventional THA cohorts [43]. Samuel et al.’s 2021 systematic review also found no significant difference in infection rates, although there is not a substantial body of evidence available to validate this [42]. Interestingly, initial reports in the AOANJRR showed increased revision rates in robotic unicompartmental knee arthroplasty for infection; however, this has shown a trend towards insignificance in the latest report [4]. Theoretically, there may be a greater infection risk with greater numbers of operating room personnel and the insertion of extra bone pins, and the robotic arm may be manoeuvred above the surgeon or their assistant’s chest, leading to compromise of sterility early in their practice. However, these occurrences have not been validated in the literature.

#### 4.3.2. Blood Loss

Reducing blood loss can hasten recovery time and hospital length of stay. Earlier data based on procedures using ROBODOC by Schulz et al. showed a higher intraoperative blood loss, but the recent trend has been equivocal [8]. With more modern systems, however, and despite a longer operation time, Bukowski et al. demonstrated a significantly reduced blood loss (374 +/− 133 mL vs. 423 +/− 186 mL) [44]. However, again with a longer operation time, Lim et al. found no significant difference between the two treatment arms (1010cc robotic vs. 895cc conventional), and this was supported by Chen et al. in their meta-analysis [15,45].

#### 4.3.3. Operative Time

Longer surgical time increases the risk of blood loss and infection. Kumar et al. showed an average operative time of 19 min longer per robotic THA than conventional [38]. However, Chen et al. showed no statistically significant difference in surgical time [15].

#### 4.3.4. Learning Curve

Whilst the learning curve to associated with implementing robotic THA into clinical practice tends to be time and efficiency related, several studies have reported on accuracy and outcome measures. Ng et al. have described a learning curve of 12–35 robotic THA procedures, and Redmond et al. demonstrated a significantly lower risk of acetabular component malpositioning and reduced operative time with increased robotic surgical experience [46,47]. Kamara et al. showed a shorter period, with competency achieved after only 10 robotic procedures, and this was reiterated by Kayani et al., with a learning curve of 12 cases [48,49]. Interestingly, in a prospective trial of 120 patients comparing newly fellowed and senior arthroplasty surgeons between both anterior and posterior approaches, Kolodychuk et al. found that use of robotic-assisted technology mitigated the learning curve for the new surgeon, as evidenced by no significant difference in radiological outcomes and operative time across both the anterior and posterior approaches to THA when compared to an experienced surgeon [50].

#### 4.3.5. Dislocation and Revision Rates

Domb et al. found no significant difference in the robotic THA revision rate in comparison to conventional THA over a five-year period; however, the suggestion was made that their more accurate cup placement leads to improved functional outcomes and enhanced prosthesis durability [40]. Samuel et al. also demonstrated equivocal revision rates between the conventional and robotic cohorts in their systematic review at varying follow-up rates between 90 days and 14 years, with zero of the seven studies reviewed showing statistical significance. Additionally, the five studies specifically comparing MAKO robotic THA and conventional THA had the same findings [42]. This comparable revision rate was reiterated by Kumar et al. in their systematic review and meta-analysis [38]. Both studies also demonstrated an equivocal dislocation rate [38,42].

#### 4.3.6. Radiation Exposure

Tarwala et al. have commented on the requirement for a preoperative CT scan, instead of the usual plain film radiographs, subjecting patients to a three-fold increase in radiation exposure [24]. However, it does remove the requirement for fluoroscopy for THA from an anterior approach, and novel scanning techniques dramatically reduce the radiation dose conferred during scanning without compromising image quality [51].

#### 4.3.7. Cost

The question also looms of the cost-efficiency of implementing new systems and technology, when compared to an already costly conventional technique. Use of a robotic THA system includes costs such as installation of the robot and its relevant software, annual service fees, training of theatre staff, equipment, sterilisation, and additional costs associated with preoperative imaging. It may be difficult to justify the high start-up costs given that the existing body of evidence has not shown any significant difference in outcomes.

With so few companies manufacturing surgical robots, start-up costs are reported as over one million dollars for the robot itself, not including implants, disposable equipment, annual servicing, and maintenance. However, development of Smith & Nephew’s CORI system adds competition to the market, and we expect to see a reduction in overall cost as a result.

Despite this, there are financial advantages to consider. Reduced complication and re-operation rates could offset the heavy start-up costs over time, as could a fewer number of trays which would minimise costs associated with sterilisation [52]. Additionally, when incorporating 90-day resource consumption, assessing costs associated with the index procedure, hospital LOS and rehabilitation, Pierce et al. showed the robotic THA to average USD $785 less than conventional, attributable to a lower requirement for rehabilitation and home nursing postoperatively [53].

Data based on the American health care system, published by Maldonado et al., found robotic THA to be cost-effective for patients both under Medicare and those privately insured. They assessed costs associated with infection, dislocation, major complications, and revision over five years, and found that overall, robotic THA saved USD 945 for Medicare and up to USD $1810 for private patients [54]. Conversely, in a statistically matched cohort, Kirchner et al. found robotic THA to have a higher associated cost when compared to conventional THA, despite a shorter hospital length of stay (LOS) [55]. Fontalis et al. suggest that implementation of robotic THA systems could be feasible if they reduce overall hospital LOS and complications, and increase implant longevity; however, the current body of evidence is inadequate [12].

The above outcomes are summarised below in Table 1.

## 5. A Look to the Future

Modelling by Sloan et al. shows primary THA rates in the United States are expected to grow 71% to 635,000 joints by 2030, and by 2060, an expected 1.23 million THA each year (a 330% increase on current numbers) [56]. Australian rates see a similar rise; Ackerman et al. predict a 208% rise in primary THA by 2030, conferring a total cost for the Australian healthcare system of AUD $5.32 billion. They identify that a population level reduction in obesity rates could amount to 8062 less primary total knee and total hip arthroplasties, with a saving of AUD $170 million to the healthcare system. These concerning numbers, with no evidence of down-trending obesity rates, demonstrate an untenable future to our healthcare system in conjunction with the expected rise in revision THA in our ageing population [57]. Almost fifty thousand THA’s were performed in Australia in 2020 alone (a 123% increase since 2003), and we can only expect this to continue to increase with the ageing “Baby Boomer” population [4,58].

It is well understood that low surgeon output and higher patient BMI contribute to inaccurate acetabular cup placement, and so the robotic system could benefit a less experienced surgeon to achieve greater accuracy. Additionally, rising worldwide obesity rates make the average patient cohort overall more difficult to accurately position implants intra-operatively, so intra-operative assessment may rectify outcomes in this cohort [59]. At present, there are no studies assessing outcomes respective to BMI, but this patient cohort may reap the greatest benefit from robotic THA.

Data pertaining to use, complications and outcomes of robotic-assisted THAs are currently not reported in the AOANJRR, nor in the registries of Canada, the United States, or the United Kingdom. However, since robotic-assisted TKA data has been reported in the AOANJRR since 2015, we should expect this data to become available in the foreseeable future [4]. With data from large joint registries, we will have greater insight into the benefits of robotic-assisted THA, if these benefits do indeed exist.

Access to teaching and education can be limited given the availability of systems to larger and higher volume arthroplasty centres. Haddad et al. suggested that ways to increase exposure include Virtual Reality (VR) simulation models, cadaveric laboratories, observation of live operations, and even interactive computer games [60].

Whilst the evidence is there that robotic THA improves radiological outcomes, it is yet to show proof that this confers improved functional outcomes. With more evidence and outcomes, we derive more avenues for improvement, and we expect to see ongoing advancements in this field in the future, including revision arthroplasty and incorporation into orthopaedic surgical training as a routine procedure, comparable to conventional THA. Much of the high-quality evidence involves both automatic and semi-active systems, so more high-quality evidence looking at each robotic system in isolation is required. With more companies entering the robotic race, we may start to see a reduction in overall costs as a result. However, given the already high satisfaction rate and evidence over long follow-up periods available with conventional THA, decisive data are required to prove their worth. As the true long-term benefit of a prosthesis is only evident after decades, further long-term studies will be required to demonstrate whether robotic-assisted total hip arthroplasty is to be the gold standard in the restoration of ideal hip biomechanics and clinical outcomes.

## Figures and Tables

**Figure 1 jcm-11-06674-f001:**
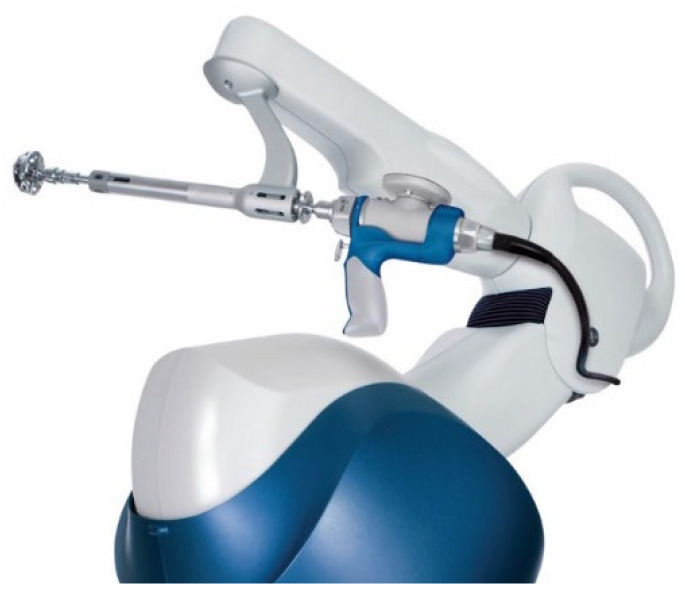
Stryker MAKO robot with acetabular reamer attachment.

**Figure 2 jcm-11-06674-f002:**
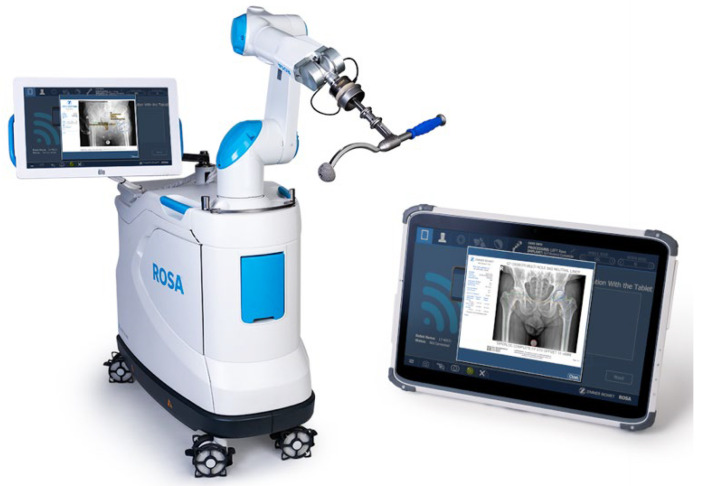
ROSA system including example of planning system.

**Figure 3 jcm-11-06674-f003:**
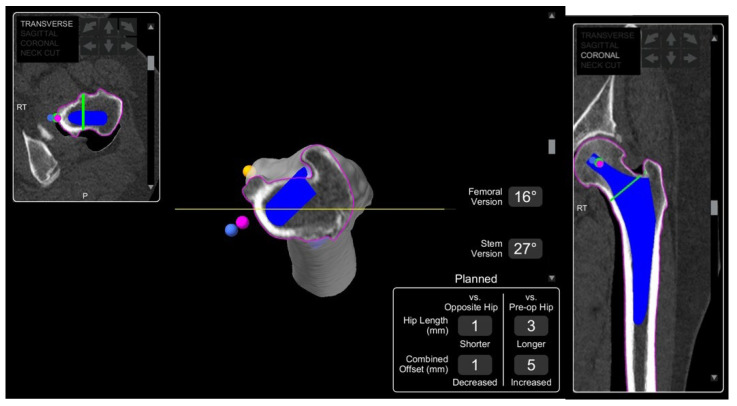
MAKO robotic-arm-assisted system planning of femoral stem insertion in coronal and transverse planes. The dark blue stem position is relative to 0 degrees (yellow horizontal line). The pink dot indicates the native centre of the femoral head, the orange dot represents the centre of the lesser trochanter, and the blue dot landmarks the selected stem head centre. The pink pine indicates the edge of the bone. These colour indications are consistent through all further pre-operative planning images below.

**Figure 4 jcm-11-06674-f004:**
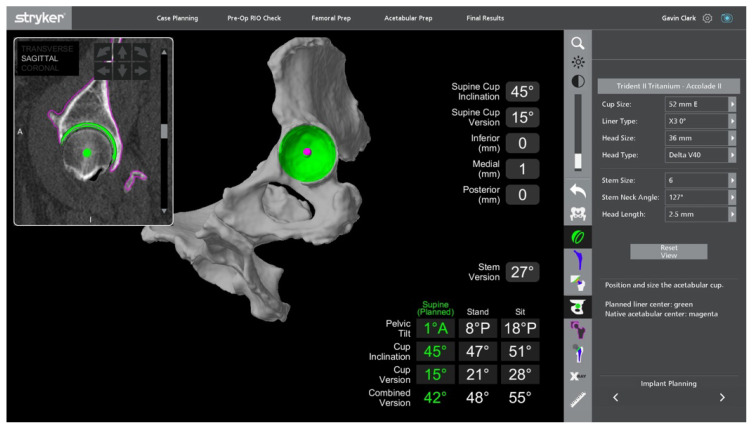
MAKO-assisted robotic THA planning cup placement. The green area is the planned bone to be removed when reaming, and the green dot indicates the planned cup centre.

**Figure 5 jcm-11-06674-f005:**
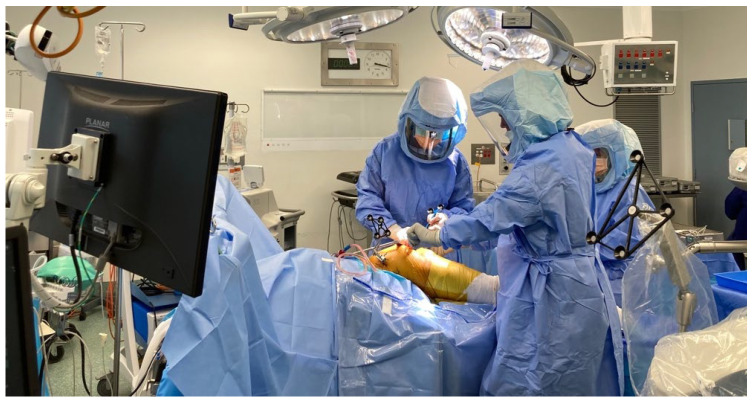
Intra-operative set-up, with screen displaying operative screen positioned conveniently for the surgeon.

**Figure 6 jcm-11-06674-f006:**
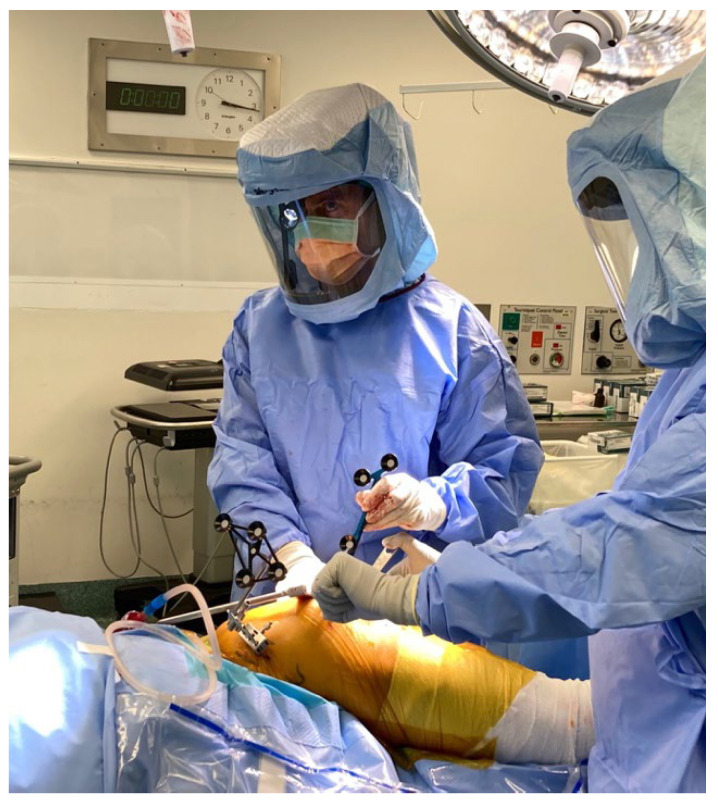
Tracking array positioned in iliac crest with hand-held array to map the acetabulum.

**Figure 7 jcm-11-06674-f007:**
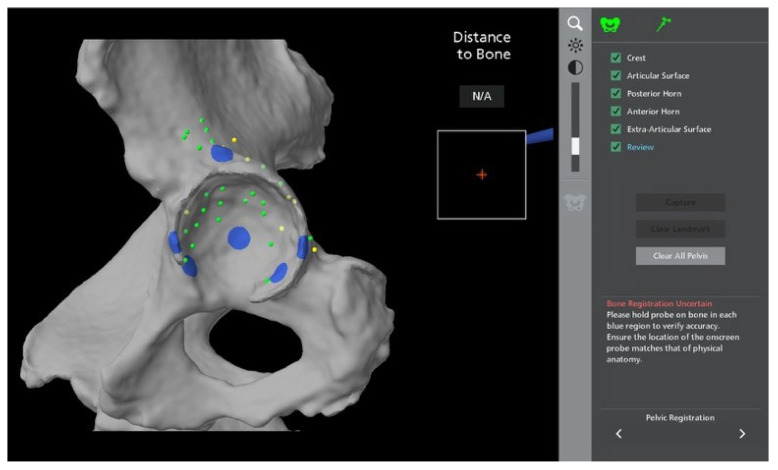
Pelvic registration. The blue areas are checkpoints to be verified with the probe, as listed on the top right of the image. The green dots indicate where the probe has registered points exactly as planned, and yellow dots indicate where the registration is slightly off plane.

**Figure 8 jcm-11-06674-f008:**
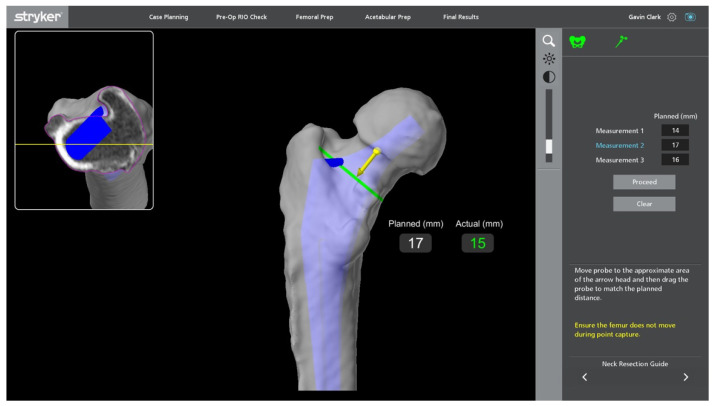
Intra-operative image templating neck resection (green line). The blue area is a registration point, and the yellow arrow is the starting point for probe registration.

**Figure 9 jcm-11-06674-f009:**
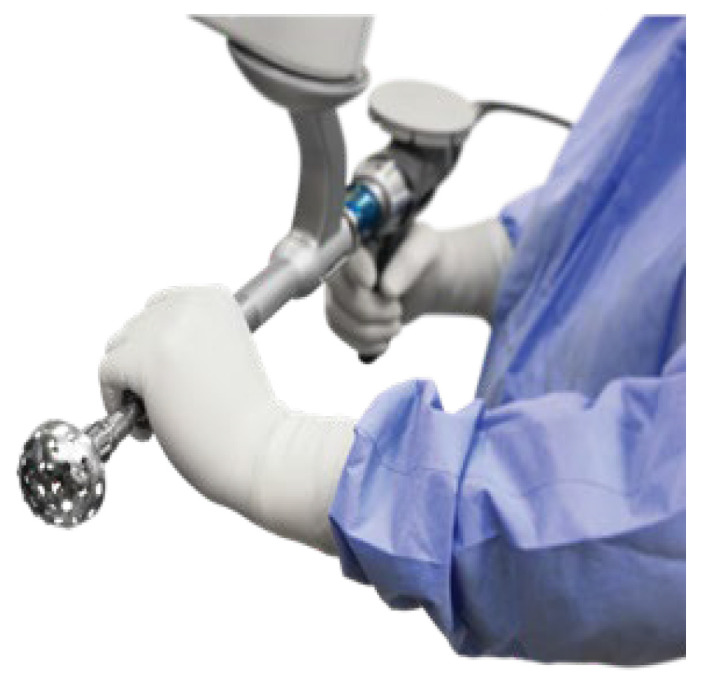
MAKO robotic arm attached to an acetabular reamer to provide feedback and guide surgical reaming of the acetabulum intraoperatively.

**Figure 10 jcm-11-06674-f010:**
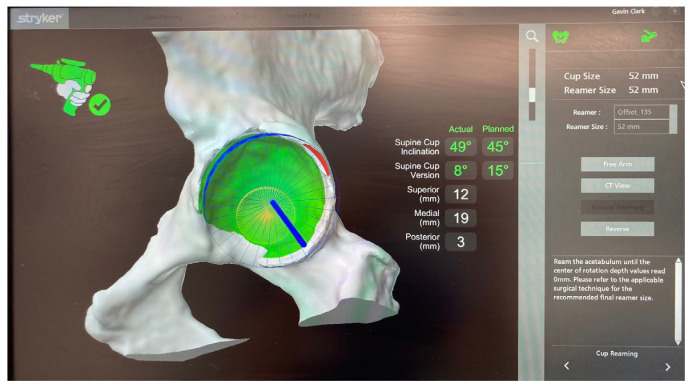
Intra-operative images detailing acetabular reaming relative to pre-operative plan. The blue areas indicate the reamer position. The green area is the area to be reamed, and the red area denotes minimal deviation from pre-operative plan (any further and the system would shut off), and the numbers in white indicate real-time distance from pre-operative plan as the operator reams.

**Figure 11 jcm-11-06674-f011:**
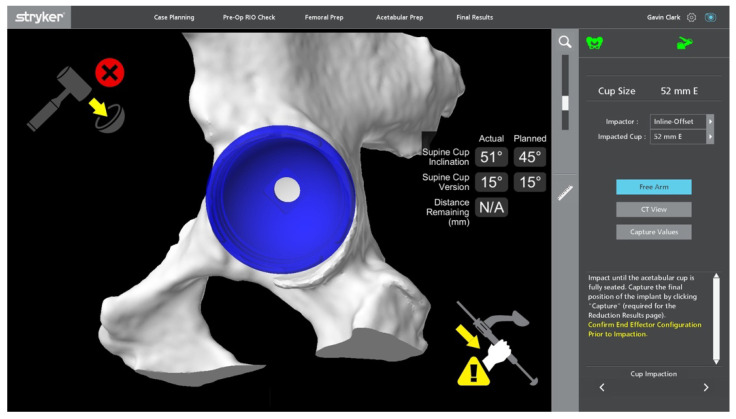
Feedback of depth of cup impaction. The red cross in the top right indicates the system is not “locked in”, and as indicated by “distance remaining”, the cup is placed in accordance with the plan.

**Figure 12 jcm-11-06674-f012:**
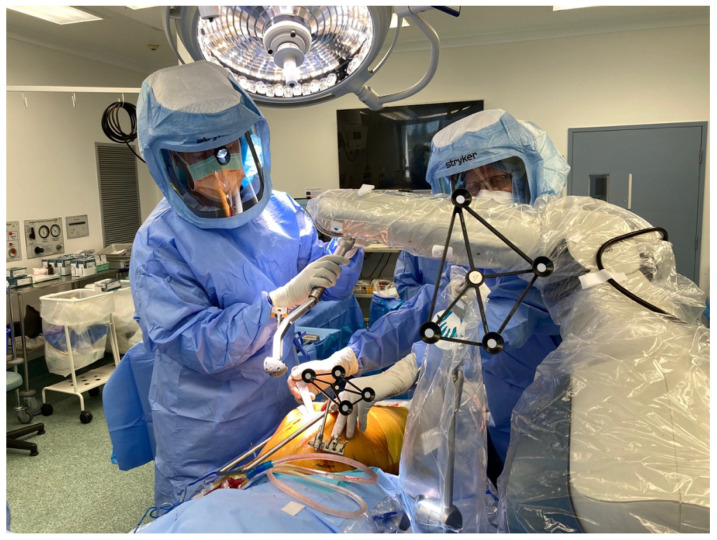
Insertion of cup using MAKO system.

**Figure 13 jcm-11-06674-f013:**
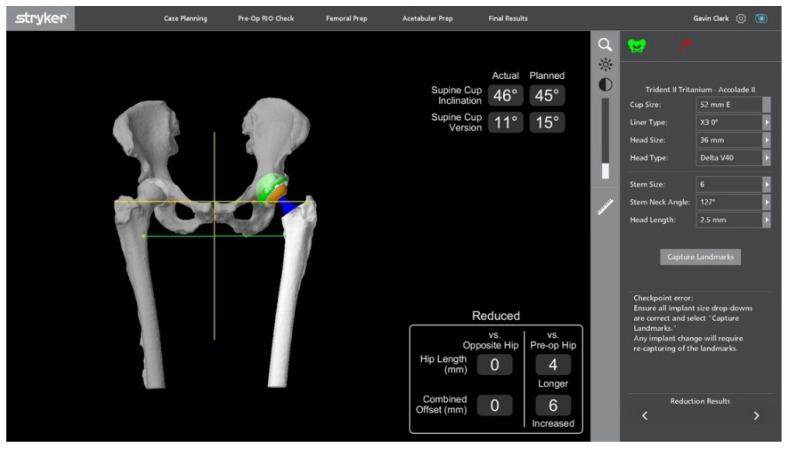
Post-reduction assessment detailing comparable leg lengths and offset to contralateral side. The yellow lines cross bilateral greater trochanters, and the green lines cross bilateral lesser trochanters, showing equivocal leg lengths.

**Table 1 jcm-11-06674-t001:** Summary of papers reviewed in this article.

Author	Year	Design	Robot Type	Evidence
Bukowski et al. [44]	2016	Retrospective	MAKO	rTHA reduced blood loss vs. cTHA (374 +/− 133 mL vs. 423 +/− 186 mL, *p* = 0.035)
Chen et al. [15]	2018	Systematic review and meta-analysis		No significant difference in surgical time rTHA vs. cTHA (however, favoured cTHA) Higher HO rate post rTHA (32/142 rTHA vs. 18/133 cTHA, *p* = 0.04)
Chen et al. [32]	2021	Review		rTHA more accurate acetabular cup placement Equivocal functional scores rTHA vs. cTHA at 5 years Equivocal blood loss rTHA vs. cTHA
Clement et al. [37]	2021	Propensity score-matched prospective	MAKO	rTHA significantly superior restoration of leg length (2.3 mm greater rTHA vs. 3.6 mm cTHA) rTHA–significantly higher OHS (2.5 points) and FJS (21.1 points) with FJS having clinical significance
Domb et al. [39]	2015	Retrospective	MAKO	Comparable LLD rates rTHA vs. cTHA (97% <10 mm, no significant difference rTHA vs. cTHA)
Domb et al. [40]	2020	Propensity score-matched retrospective	MAKO	rTHA higher HHS, FJS and VR-12 Physical (all significant) No significant difference in revision rates over 5 years
Emara et al. [31]	2021	Systematic review and meta-analysis		rTHA superior acetabular cup positioning (significant) rTHA significantly lower LLD vs. cTHA (−0.33 mm vs. −1.24 mm)
Han et al. [33]	2019	Systematic review and meta-analysis		No significant difference in development of HO rTHA vs. cTHA No difference rTHA vs. cTHA in HHS, WOMAC, Merle D’Aubigne scores post-operatively (none reaching significance)
Illgen et al. [43]	2017	Retrospective	MAKO	rTHA vs. cTHA no difference in infection rates
Kamara et al. [48]	2017	Retrospective	MAKO	rTHA competency achieved after 10 procedures
Kayani et al. [49]	2021	Prospective	MAKO	rTHA competency achieved after 12 procedures
Kirchner et al. [55]	2021	Retrospective	Not specified	rTHA higher cost (USD $20,046 vs. cTHA USD $18,258), despite shorter hospital LOS
Kolodychuk et al. [50]	2021	Prospective	Not specified	rTHA mitigated learning curve, with no significant difference in radiological outcomes and operative time between new and experienced surgeons
Kumar et al. [38]	2021	Systematic review and meta-analysis		rTHA reduced LLD vs. cTHA (mean difference 1.44 mm, *p* = 0.01) rTHA longer operative time (mean difference 19.48 min, *p* = 0.02) No significant difference in dislocation and revision rates rTHA vs. cTHA
Lim et al. [45]	2015	Prospective	ROBODOC	No significant difference in blood loss (rTHA 1010cc vs. cTHA 895cc)
Maldonado et al. [54]	2021	Computer simulation		rTHA significant cost reduction, saving USD 945 per public patient, and USD 1810 for private patients
Ng et al. [46]	2021	Systematic review and meta-analysis		Learning curve to rTHA competency 12–35 patients
Nishihara et al. [41]	2006	Prospective	ORTHODOC ISS	Equivocal time to walking 500 m, rTHA > cTHA number of patients walking 6 blocks in 13 days (significant) Equivocal Merle d’Aubigne hip score 3 months post-operatively (rTHA 15.8 vs. cTHA 15.3, insignificant), rTHA significantly improved scores on the same scale at 2 years (rTHA 17.4 vs. 17.1)
Pierce et al. [53]	2022	Propensity score-matched retrospective	Not specified	rTHA overall lower 90-day cost (assessing index procedure, hospital LOS and rehabilitation) averaging USD 785 less per patient.
Redmond et al. [47]	2015	Retrospective	MAKO	Significantly lower risk of malpositioned acetabular cup (103/105 in Lewinnek’s safe zone, and 99/105 in Callanan’s safe zone) and a shorter operating time with the final 70 rTHA cases, which reached significance.
Samuel et al. [42]	2021	Systematic review		No significant difference in functional outcomes, and between MAKO and ROBODOC No significant difference in infection rates Equivocal dislocation and revision rates rTHA vs. cTHA
Schulz et al. [8]	2007	Prospective	ROBODOC	rTHA higher intraoperative blood loss and transfusion requirement
Tarwala et al. [24]	2011	Review		rTHA 3x increase in radiation exposure due to pre-operative planning CT scan

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
