# Peer review of "Robotics in Total Hip Arthroplasty: Current Concepts"

_jcm, 2022, doi:10.3390/jcm11226674_

Round 1
Reviewer 1 Report
Thanks for the opportunity to review this article. The purpose of the narrative review is to summarize the evidence in the literature on the use of robotics in hip replacement.
In my opinion, the literature can benefit from a summary of the evidence on this topic, and reading this review can provide a quick update and a starting point for more in-depth evaluations.
The article is well written and organized. The points covered in the discussion are relevant and interesting. however, I believe that further effort in describing the systems in use today is necessary (I am referring to "how it works") and that relevant figures of the phases of preparation and use of the major systems are required to aid understanding.
Thank you.
Author Response
Reviewer #1
We have incorporated an image from Zimmer detailing the surgical process for their ROSA system, as suggested. Unfortunately, the other systems mentioned have minimal evidence available, and were not detailed to maintain simplicity of the article.
Reviewer 2 Report
Thank you for your hard work in organizing this vast amount of content.
1. The history, current position, and methods of robots in implementing THA. You described the comparison with the conventional op. well. This paper is a review article and needs more figures and tables to help readers understand. It would be better if the papers that published the results of robotic surgery were collected and organized into a single "table" for comparison.
2. Because I have experienced the MAKO system, I can understand it well, but readers who only encounter it as a paper may not understand it well. Insert a figure for each surgical procedure.
Author Response
Reviewer # 2
It would be better if the papers that published the results of robotic surgery were collected and organized into a single "table" for comparison.
Please see table as recommended. Line 467.
Because I have experienced the MAKO system, I can understand it well, but readers who only encounter it as a paper may not understand it well. Insert a figure for each surgical procedure.
Please see included clinical images and workflow screens for stages of the procedure.

Reviewer 3 Report
Your article is interesting and clear, but the review is also based on articles with poor case studies. The article also does not give us any new information that could enrich literature (there are similar works as proposed). From a conceptual point of view, the inclusion of the Navio is not adequate being a computer navigator and not a Robot, although the introduction in the first part is correct. The article to be adequate in my opinion need for conceptual corrections, where it decreases the reference to past systems robots and focuses on the results of current systems, with a comparison between past and present.
Major revisions to editorial standards are necessary in my opinion. Once these revisions are made, the article will have considerable potential.
Author Response
From a conceptual point of view, the inclusion of the Navio is not adequate being a computer navigator and not a Robot, although the introduction in the first part is correct.
We have removed reference to the NAVIO system. We have updated this to mention the Smith & Nephew CORI system, although this system is still in pilot-phase development.
Line 88, line 436.
The article to be adequate in my opinion need for conceptual corrections, where it decreases the reference to past systems robots and focuses on the results of current systems, with a comparison between past and present.
Most of the systematic reviews that were reviewed included RCTs that compared both MAKO and ROBODOC systems with conventional techniques, and few compared the two systems directly on a large scale. Whilst there is an increasing number of RCTs assessing more modern systems, there is still a paucity of high-quality, large-volume data in this area.

Round 2
Reviewer 1 Report
The authors have addressed all may concerns. Thank you.